# Revisiting Regulated Cell Death Responses in Viral Infections

**DOI:** 10.3390/ijms23137023

**Published:** 2022-06-24

**Authors:** Devasahayam Arokia Balaya Rex, Thottethodi Subrahmanya Keshava Prasad, Richard K. Kandasamy

**Affiliations:** 1Centre for Integrative Omics Data Science, Yenepoya (Deemed to be University), Mangalore 575018, India; rexprem@yenepoya.edu.in; 2Center for Systems Biology and Molecular Medicine, Yenepoya Research Centre, Yenepoya (Deemed to be University), Mangalore 575018, India; 3Centre of Molecular Inflammation Research (CEMIR), Department of Clinical and Molecular Medicine (IKOM), Norwegian University of Science and Technology, 7491 Trondheim, Norway; 4College of Medicine, Mohammed Bin Rashid University of Medicine and Health Sciences, Dubai P.O Box 505055, United Arab Emirates

**Keywords:** cell death, death receptors, viral infection, signaling, regulated cell death

## Abstract

The fate of a viral infection in the host begins with various types of cellular responses, such as abortive, productive, latent, and destructive infections. Apoptosis, necroptosis, and pyroptosis are the three major types of regulated cell death mechanisms that play critical roles in viral infection response. Cell shrinkage, nuclear condensation, bleb formation, and retained membrane integrity are all signs of osmotic imbalance-driven cytoplasmic swelling and early membrane damage in necroptosis and pyroptosis. Caspase-driven apoptotic cell demise is considered in many circumstances as an anti-inflammatory, and some pathogens hijack the cell death signaling routes to initiate a targeted attack against the host. In this review, the selected mechanisms by which viruses interfere with cell death were discussed in-depth and were illustrated by compiling the general principles and cellular signaling mechanisms of virus–host-specific molecule interactions.

## 1. Introduction

Programmed cell death (PCD) or apoptosis is an important form of cell-autonomous immune control over intracellular pathogens [1,2]. PCD is mainly responsible for regulating animal development and tissue homeostasis, which regularly occurs in a broad range of human diseases, including immunological, developmental problems, neurodegeneration, and cancer [3,4,5,6]. The known cell death pathways associated with viral infections include apoptosis, necroptosis or programmed necrosis and pyroptosis/pyronecrosis [7]. Regulated cell death (RCD) is a critical, unavoidable part of life and a common cellular response to viral attacks [8,9]. There are many types of RCD reported so far, and they can be either non-inflammatory or pro-inflammatory [10,11]. Lesser-known non-apoptotic RCD associated with viral infection are ferroptosis, parthanatos [12,13], entosis [14], mitochondrial permeability transition (MPT)-dependent necrosis, pyroptosis, pyronecrosis [15,16] and anoikis [17]. NETosis or Etosis (neutrophil extracellular traps formation) plays a crucial role in host immune responses, where neutrophil granulocytes are activated through immune regulatory functions such as (i) phagocytosis, (ii) formation of reactive oxygen species (ROS), (iii) degranulation, and (iv) generation of neutrophil extracellular traps (NETs) [18,19,20], alkaliptosis (a pH-dependent form of regulated non-apoptotic cell death) [21], and oxeiptosis (ROS-induced caspase-independent apoptosis) [22,23].

In response to viral infection, many cells undergo apoptosis, resulting in a decrease in the release of progeny virus. Viruses have thus evolved a variety of mechanisms for modulating host cell apoptosis. Viruses can interfere with either the highly conserved ‘effector’ mechanisms of PCD or mammalian-specific regulatory mechanisms. Aside from providing a selective advantage to the virus, the ability to prevent apoptosis is critical in the transformation of the host cell by oncogenic viruses. This article provides a focused review of apoptosis, necrosis, pyroptosis, ferroptosis, entosis, methuosis, parthanatos and demonstrates how research on viruses and viral infections has aided our understanding the mechanism of PCD. We also discussed virus-induced signaling mechanisms in both the well-known and lesser-known RCD.

## 2. Major Cell Death Pathways

The Nomenclature Committee on Cell Death (NCCD) has developed guidelines for the morphological, biochemical, and functional definitions and interpretations of cell death. The NCCD’s mission is to provide a widely accepted nomenclature for cell death in order to support the field’s continued development. Various programs of RCD were described, and their research continues to progress. Accordingly, it defined terms based on biological functions such as intrinsic apoptosis, extrinsic apoptosis, and MPT-driven necrosis and pyroptosis from a molecular perspective [24,25,26] (Figure 1).

### 2.1. Apoptosis

Apoptosis is a naturally occurring process that plays a distinct role in morphological and biochemical changes. It is also called PCD and can occur through two distinct apoptotic pathways: the intrinsic and extrinsic pathways [27]. Some of the main characteristics of apoptotic cells include chromatin condensation, DNA fragmentation, phosphatidylserine (PtdSer) cell surface exposure on the outer leaflet of the plasma membrane, and the formation of apoptotic bodies [28]. The key mediators of apoptosis are caspase enzymes [29]. Apoptosis is characterized morphologically by membrane blebbing, chromatin condensation, intra-nucleosomal DNA fragmentation, and the formation of apoptotic bodies and is mediated by a specific cysteine protease family of proteins called caspases [30]. Cytochrome c (Cyt c) is found in the inner membrane areas of healthy mitochondria, where it serves as an electron shuttle in the respiratory chain to interact with cardiolipin (CL) [31,32]. Several proapoptotic stimuli cause the outer membrane to permeabilize, allowing contact between the intermembrane and intercristae gaps and the mobilization of Cyt c from CL, allowing Cyt c release [33]. Cyt c facilitates the allosteric stimulation of apoptosis-protease activating factor-1 (Apaf-1), which is necessary for caspase-9 and caspase-3 proteolytic maturation in the cytosol. Mitochondrial disruption triggers the release of Cyt c, which forms apoptosomes with Apaf-1 and caspase-9 during apoptosis. Dimerization activates caspases in apoptosomes, resulting in caspase signaling. Through the mitochondrion, apoptosis-inducing factor and endonuclease G may also promote caspase-9-independent apoptosis [34].

Caspases that were activated cause apoptotic cells to be dismantled. However, cytosolic Cyt c was linked to important cell processes such as differentiation, implying that its release is not always all-or-nothing and that mitochondrial outer membrane permeabilization does not always result in cell death [32]. Apoptosis is triggered when cell-surface death receptors such as Fas (also known as CD95) are bound by their ligands (the intrinsic pathway) or when pro-apoptotic proteins of the Bcl-2 family cause the permeabilization of the mitochondrial outer membrane (the extrinsic pathway). The activation of both pathways results in the activation of the caspase family of proteases, ultimately resulting in the cell’s destruction [35]. To determine the signaling processes that occur in apoptosis, Chen et al. demonstrated that mitochondrial membrane potential was lost when caspase-9 was dimerized, and anti-apoptotic Bcl-2, Bcl-xL, and Mcl-1 were cleaved. Cleavage-resistant of Bcl-2, Bcl-xL, or Mcl-1 also significantly reduced caspase-9-dependent Cyt c release and mitochondrial membrane potential loss [36].

Natural killer (NK) cells are an immune effector cell population with an intrinsic ability to kill virus-infected and tumor cells without prior antigen sensitization. NK cell-mediated cytotoxicity is regulated by integrating signals from a repertoire of activating and inhibitory receptors. Activated NK cells can kill target cells through different mechanisms, including the release of preformed cytolytic granules and pro-inflammatory cytokines as well as trigging cell apoptosis through death-receptor pathways by inducing death ligands such as the Fas ligand and the tumor necrosis factor-related apoptosis-inducing ligand (TRAIL) [37]. In recent investigations, TRAIL has been a useful biomarker for distinguishing between bacterial and viral infections [38]. High levels of interferons (IFNs) were related to lymphocyte death, and TRAIL and its receptor, death receptor 5 (DR5), were suggested as possible molecules [39]. The TRAIL and DR5 expression levels in patient blood samples [40] and IFN alpha/beta expression by plasmacytoid dendritic cells (pDCs) in tonsil tissue are significantly higher in progressors than non-progressors [41].

### 2.2. Necroptosis

Necroptosis or programmed necrosis is considered a “trap door” to eliminate pathogens when the caspase system gets impaired [42]. Morphologically, necroptosis is characterized by increased cell volume, membrane rupture and swelling of organelles, cellular collapse, and release of cellular contents [43]. The hallmarks of necrotizing non-apoptotic cell death are regulated necrosis (namely necroptosis) and rapid loss of plasma membrane integrity [44]. Necrosis is regulated by several mediators, including death receptors (DRs) such as tumor necrosis factor receptor-1 (TNFR1), CD95, and DR4/5, IFNs, Toll-like receptors (TLRs), intracellular RNA and DNA sensors. This necrosis is caused by the protein receptor-interacting protein kinase-3 (RIPK3) and mixed lineage kinase domain-like (MLKL) [45]. Recent evidence reveals that necroptosis is associated with RIPK3 and MLKL-induced inflammation [46] and is implicated in the pathogenesis of several diseases such as strokes [47], myocardial infarction [48], and pulmonary disease [49] and host defense against the virus [50]. Activated RIPL3 and MLKL complexes are the key necrosis regulators that orchestrate tissue inflammation and injury. A recent study reported that phosphorylated RIPK3 and MLKL signaling promotes inflammation, airway remodeling, and emphysema in chronic obstructive pulmonary disease (COPD) in the lung tissue of COPD patients [51]. Several other factors are also mainly involved in this mode of cell death. For example, intracellular ATP depletion [52,53], lysosomal enzyme release [54], and excessive ROS generation determine the cell death fate by necrosis [55].

### 2.3. Pyroptosis

Pyroptosis is a newly discovered programmed cell death, and it was studied in the context of cancer [56] and neuronal diseases [57]. Morphologically, pyroptosis greatly differs from other cell death mechanisms such as apoptosis and necrosis in terms of occurrence and regulatory mechanisms [58,59,60]. Pyroptosis differs from apoptosis by being highly pro-inflammatory, involving the formation of membrane pores, is lytic and is driven by caspase-1, previously regarded as the “IL-1β-converting enzyme”, which is not associated with apoptosis. Moreover, it differs from other pore-forming RCDs, such as necroptosis, by involving the gasdermin (GSDM) family of proteins as the membrane pore-forming agent [24]. It manifests by activation of one or more caspases, primarily caspase-1 in humans and mice, human caspase-4/5 in humans and caspase-11 in mice [61,62] and thus can broadly be classified into canonical caspase-1 dependent and non-canonical caspase-1 independent pyroptosis. Both the types, however, remain morphologically similar.

#### 2.3.1. Caspase-1-Dependent Pyroptosis

Caspase-1 occurs as an inactive 45 kDa precursor protein in the cytosol. Multimeric protein complexes within the cytosol, known as inflammasomes, are activated in response to pathogen-associated molecular patterns (PAMPs) or danger-associated molecular patterns (DAMPs) [63] generated by microbial infection and endogenous stress [64]. Inflammasomes can result in either cytokine processing or pyroptosis. An inflammasome is composed of caspase-1 subfamily caspases such as caspases-1/4/5 in humans and caspases-1/11 in mice. These inflammasome-initiating sensors include NOD-like receptors (NLRP1, NLRP3, NLRC4), IFN-inducible protein AIM2 is also known as absent in melanoma 2 (AIM2) or pyrin, with or without the apoptosis-associated speck-like protein containing a CARD (ASC), an inflammasome adaptor protein [65,66,67]. All three classes of inflammasome sensors and ASC contain CARD (caspase activation and recruitment domains), pyrin domains (PYD), or both. NLRs recognize those danger signals introduced into the host cell cytosol, resulting in a massive release of inflammatory cytokines (IL-1 β, IL-18). Some NLRs, such as NLRC4 [68], can directly bind to caspase-1, while others, including NLRP3, bind to the adapter protein ASC, which contains a CARD domain that interacts and activates with caspase-1, including NLRP3 [69]. Caspase-1 is activated within this complex via dimerization and autoproteolysis [70]. A single NLR [71,72] or multiple NLRs [73] can activate caspase-1. Once activated, caspase-1 also produces plasma-membrane pores of 1.1–2.4 nm in diameter, causing water influx, the release of cellular ionic gradients, cell swelling, and osmotic lysis [74]. The mechanism of the pore formation during pyroptosis, which is still less known, is attributed to the action of a newly discovered component of inflammasomes, gasdermin D (GSDMD) [75]. The pyroptosis execution protein GSDMD is activated by inflammatory caspases (caspase-1/11/4/5), which cleaves GSDMD to expose its N-terminal domain for relocation to the plasma membrane, producing large oligomeric pores [76,77,78,79]. Caspase-1 activates the pro-forms of IL-1β and IL-18 and releases them through the GSDMD pores during pyroptosis, although the exact secretion mechanism remains unclear. The steps that lead to cell death after activation of caspase-1 mediated signaling is not fully understood.

#### 2.3.2. Caspase-1-Independent Pyroptosis

Caspase-1-independent pyroptosis is considered non-canonical pyroptosis and is activated by apical activators of the inflammasome, including caspases-4/5/11 [80,81,82,83]. Unlike canonical pyroptosis, non-canonical pyroptosis is driven by non-canonical inflammasomes without any cytokine release. However, caspase-1-dependent pyroptosis has reported a high affinity for GSDMD, IL-1β, and IL-18, whereas caspase-1-independent pyroptosis has a comparable affinity for GSDMD, which is less likely to release IL-1β and IL-18 cytokine [75,84]. Activated caspase-11 causes cell swelling, cell membrane damage, free radical production, and inflammation, which are the morphological characteristics of necroptosis. Caspase-11 also turns procaspase-1 into caspase-1 family and caspase-3, which leads to a massive release of inflammatory cytokines such as IL-1β and IL-18 [85]. Additionally, there is exciting evidence that there is a crosstalk between non-canonical and canonical pyroptosis, in which non-canonical inflammasome-mediated activation of GSDMD triggers canonical inflammasome activation and cytokine release [86]. The activation of GSDMD by non-canonical inflammasomes leads to canonical inflammasome activation and cytokine release in the intracellular DAMP generation [87].

## 3. Cell Death Signaling in DNA Virus Infection

Cell death is an integral part of the host–pathogen defense mechanism against an invading intracellular pathogen such as viruses. It can occur at any stage of viral infection, including virus attachment to host cell receptors, virus entry into cells, and viral nucleic acid integration into the host genome [88,89]. Apoptosis, necroptosis, and pyroptosis are the three crucial cell death mechanisms limiting virus replication in the infected host cell and simultaneously inducing pro-inflammatory and innate immune responses against the viral infection [90]. However, in some cases, the viral infection does not result in cell death but rather enhances broader virus dissemination, and it is involved in causing tissue damage, which enhances the viral infection [91]. Over time, viruses have evolved multiple mechanisms to modulate these host cell death mechanisms to promote their long-term survival in the host cell and spread to surrounding host cells [88]. Viruses require cells to be alive to replicate their genomes and generate progeny, so they have evolved a number of mechanisms to control host cell death [92]. The diverse array of DNA viruses, including Poxviridae (vaccinia virus and myxoma virus), Herpesviridae (HSV-1, Human cytomegalovirus and varicella-zoster), Adenoviridae (adenovirus type-5) and murine cytomegalovirus (MCMV) infection triggers apoptosis, pyroptosis and necroptosis, which are the three forms of cell death commonly observed in response to viral infection [7,93,94] (Figure 2).

### 3.1. DNA Viruses Mediated Apoptosis

Most of the cells undergo apoptosis upon viral infection, which reduces the release of progeny viruses [92]. Apoptosis is an imperative mechanism involved in eliminating virus-infected cells from the host. Most viruses have evolved mechanisms to inhibit or delay apoptosis, thus allowing them time to replicate and assemble viral particles. This leads to persistent infection. Many viral proteins are known to modulate apoptosis both positively and negatively. It is considered a crucial defense mechanism against viral infection, limiting viral replication, and spread. Apoptosis occurs when cell-surface death receptors, such as Fas, bind their ligands (the intrinsic pathway), or when Bcl-2-family pro-apoptotic proteins cause permeabilization of the mitochondrial outer membrane (the extrinsic pathway). Both pathways lead to the activation of the caspase protease family, responsible for the cell’s eventual dismantling [35].

Viruses modulate host cell apoptosis either by interfering with the conserved effector mechanisms of apoptosis or by interfering with regulatory mechanisms specific to host cells [95]. These distinct mechanisms can confer a selective survival advantage to the virus or the host cell in two ways: (i) preventing the cell from dying prematurely during lytic virus infection, which can result in increased release of progeny virions [95]; and (ii) promoting host cell transformation and maintenance during latent infections. Many DNA viruses encode apoptosis inhibitors, thereby facilitating their replication and persistence in the infected host cell [89]. Sometimes, viruses can also induce apoptosis and then use the remnants of host cells as a vehicle for viral transmission, avoiding recognition by the host immune system [96]. Of particular interest, Choi et al. and Zhang et al. reported that the Epstein–Barr virus (EBV) infection has a substantial role in inhibiting and translating the apoptosis inhibitor baculoviral IAP repeat-containing protein 6 (BIRC6, also known as BRUCE) and promoting apoptosis in AGS, an EBV-negative gastric cancer (GC) cell line [97,98]. In addition, DNA viruses encode inhibitory proteins such as inhibitors of apoptosis proteins (IAPs). These IAPs prevent apoptosis by interacting with and stabilizing the function of host IAPs [99]. Inhibiting DNA synthesis is a common therapeutic strategy for hyperproliferative diseases, such as viral infections, autoimmune diseases, and cancer [100]. One group of viral proteins are well-known inhibitors that target caspases; viral proteins directly bind to host protein caspases and inhibit their activity. DNA viruses such as herpesvirus family members, HSV, and rodent cytomegalovirus encode caspase inhibitors, such as ribonucleotide reductase R1 viral inhibitor of caspase-8 activation (vICA). These viral proteins inhibit CD95 death-inducing signaling complex-mediated apoptosis by interacting with the prodomain of procaspase-8 via their death effector domain, protecting cells from TNF and FasL-induced apoptosis and preventing apoptosis caspase activation [101,102].

During EBV infection, EBV nuclear antigen 1 (EBNA1) is mainly involved in suppressing NK cell responses and cell death in the newly infected peripheral blood cells. This is achieved by downregulation of unique long 16 (UL16) binding protein 1 (ULBP1) and retinoic acid early transcript 1G protein, RAET1G (ULBP5), the natural killer group 2D (NKG2D) expression along with the modulation of c-Myc expression. Thus, EBNA1 of the EBV virus performs an important role in reducing apoptosis and host immune response, thereby helping the EBV-infected cells to avoid the host immune surveillance mechanisms and apoptosis which helps the virus to persist in the host for the entire host lifetime [103]. Other DNA viral inhibitors can also block the extrinsic apoptotic pathway by inhibiting the protein called c-FLIP, which binds to FADD, or by neutralizing death ligands that promote the activation of caspase-8 [27]. They also express proteins that inhibit the intrinsic pathway of apoptosis by mimicking host anti-apoptotic proteins, such as Bcl-2, or by inactivating the function of proapoptotic Bcl-2 family members, including Bax [104]. Apoptosis can also be helpful for the virus by promoting the release of the virus from the infected host cell and allowing the spread of the virus from one infected cell to another. Proapoptotic signals could also be beneficial for viruses to blunt their immune responses [95].

### 3.2. DNA Viruses Mediated Pyroptosis

Compared to apoptosis and necroptosis, pyroptosis appears less common following virus infection [105]. Pyroptosis is a lytic PCD that serves as an innate immune mechanism to aid in the defense of the host against DNA viruses. Pyroptosis also facilitates the elimination of infected cells by limiting the survival and proliferation of intracellular pathogens [106]. The diverse array of DNA viruses activates cytosolic pattern recognition inflammasome receptors (NLRP3 and AIM2) during the infections [107,108]. DNA viruses have evolved mechanisms to modulate pyroptosis via inflammasome signaling during viral infection [107]. The pathogen sensors are an integral part of inflammasomes, and there are different types of inflammasomes depending on the pathogen sensor [109]. The NLRP3 is the most widely studied and characterized NLR inflammasome, activated during viral infection with highly diverse viruses [64,110,111]. The NLRP3 inflammasome is thought to respond to a change in cellular homeostasis and ionic balance rather than a specific microbial or viral ligand [112,113]. Over and above established microbial settings, NLRP3 contributes to immunopathology and innate and adaptive immunity quality during viral infection [114]. Caspase-1 plays a major role in the pyroptosis mechanism and inflammasomes. It regulates cell membrane integrity and an inflammatory form of lytic cell death in pyroptosis [107,108]. The AIM2 inflammasome initialization is required to activate host defense against cytosolic bacteria and DNA viruses in Aim2-deficient mice. Among DNA viruses, the AIM2 inflammasome is activated in response to the vaccinia virus (VV) and MCMV. The VV induces IL-1β release and maturation, which gets attenuated in macrophages lacking Aim2, whereas IL-18 serum levels are reduced in MCMV-infected Aim2 deficient mice [115]. More studies are required to understand the mechanism of AIM2 induction of inflammasome formation/activation and the subsequent downstream signaling.

In particular, HSV-1 is a highly prevalent double-stranded DNA virus in humans [116]. The HSV-1 is one of eight human herpesviruses (HHV), such as HSV-2, varicella-zoster virus (VZV; HHV-3), Epstein–Barr virus (HHV-4), cytomegalovirus (HHV-5), HHV-6, HHV-7, and HHV-8. The herpesviruses are large, well-adapted to human infection as they establish lifelong infection, rarely cause the host’s death, and are readily spread between individuals. However, it causes severe inflammation called herpes simplex encephalitis (HSE) [117]. A recent study by Hu et al. reported that HSV-1 induces GSDMD pyroptosis by activation of NLRP3 inflammasomes, which induces the production of IL-1β expression and caspase-1 release in mouse microglia [118]. Karaba et al. observed that HSV-1 is capable of activation of the non-canonical inflammasome without NLRP3, ASC, and caspase-1, which results in the activation of IL-1β but not IL-18 in pr-inflammatory human THP-1 macrophages [119].

Pyroptosis is a type of cell death mechanism which is often a part of the body’s response to viral infection. A study by Orzalli et al. demonstrated inhibition of protein synthesis during viral infection to be a virulence strategy used by vesicular stomatitis virus (VSV) and HSV-1 mutant virus that lacked the immediate-early protein ICP27, thus leading to pyroptotic cell death in human keratinocytes. During VSV and HSV-1 infection, leads to caspase-3 dependent cleavage of GSDME-dependent pyroptosis mechanism activation as well as a reduction in IL-1 dependent responses was observed in keratinocytes. The viral infection led to inhibition of protein synthesis which triggered pyroptosis since there was a reduction in MCL-1 protein abundance and inactivation of BCL-xL [120]. Ho Xiao et al. also reported that HSV-1, a DNA virus, triggers pyroptosis dependent on gasdermin D (GSDMD) activation through NLRP3 inflammasomes in microglial cells. This leads to the production of mature IL-1β, an inflammatory cytokine, and activation and release of caspase-1 resulting in host cell death. This reveals that inflammasome activation and GSDMD-dependent pyroptosis are important mechanisms of HSV-1-induced pathogenesis and inflammation [118].

### 3.3. DNA Viruses Mediated Necroptosis

In the case of herpesviruses, both HSV-1 and HSV-2 are shown to attenuate TNF-induced necroptosis in virus-infected human cells by interrupting receptor-interacting protein (RIP) homotypic interaction motif (RHIM) dependent interactions between RIP1 and RIP3 [121]. The HSV R1 proteins such as HSV1 ICP6 and HSV2 ICP10, contain the RHIM domains, which are the central players in inhibiting necroptosis [121,122]. These HSV R1 proteins are anti-apoptotic proteins, and also interfere with necroptosis via the RHIM-dependent interaction of RIP3 in a species-specific manner. In the human host, HSV R1 binding to RIP3 or/and RIP1 prevents the formation of the RIP1-RIP3 necrosome and prevents the activation of RIP3. This leads to inhibition of necroptosis, promoting cell survival during HSV infection and thus facilitates sustainable viral replication [121]. On the other hand, in the mouse model, the opposite effect occurs; as such, the interaction of R1 with RIP3 results in the activation of RIP3 and necroptosis, thereby limiting viral replication [123].

The third group of DNA viruses known to impair necroptosis is the gamma-herpesvirus EBV in a mechanism distinct from the previous RHIM-dependent pathway. A recent study on nasopharyngeal carcinoma cells showed that the EBV-encoded anti-necroptotic protein is the latent membrane protein 1 (LMP1), lacking an RHIM domain and using an epigenetic approach to inhibit necroptosis [124]. The LMP1, despite lacking an RHIM domain, shows interaction with both RIPK1 and RIPK3 through its C-terminal cytoplasmic tail, which consists of three C-terminal activating regions (CTARs) and CTAR2 in particular. Liu et al. demonstrated that LMP1 promotes K48- and K63-linked polyubiquitination of RIPK1 while inhibiting K63-linked polyubiquitination of RIPK3. The K63-polyubiquitinated RIPK1 serves as a docking site on the plasma membrane to activate NF-kB. This leads to the necrosome formation and switching to the cell survival state. The K48-linked polyubiquitination, on the other hand, shortens RIPK1 half-life as K48-linked polyubiquitination mainly signals protein for proteasomal degradation [125]. Another finding by Shi et al. revealed hypermethylation of the RIP3 promoter suppressing RIP3 expression as another possible mechanism for necroptosis inhibition by EBV [126]. Majorly, Z-DNA-binding protein *1* (ZBP*1*/DAI) contributes to innate host defense against viruses by triggering necroptosis. A study by Koehler et al. observed that during VV infection, the viral E3 protein competes with this Z-form RNA through an N-terminal Z-α domain and inhibits the initiation of necroptosis by preventing the death signaling. In absence of this E3 domain, there is an accumulation of Z-form RNA during the early stages of VV infection which triggers ZBP1 to recruit receptor-interacting protein kinase RIPK3 and initiate the process of necroptotic cell death. Overall, this study highlights the importance of the Z-form RNA generated during VV infection as a potential pathogen-associated molecular pattern (PAMP) and unleash the ZBP1/RIPK3/MLKL mediated necroptotic cell death [127].

Among poxviruses, VV is reported to inhibit DAI-mediated necroptosis while showing susceptibility towards TNF- or RIPK1-mediated necroptosis. The VV immune evasion gene, E3L encodes for an innate immune evasion protein, E3, with the N-terminus Z-form nucleic acid binding (Zα) domain competing with DAI to prevent DAI-dependent activation of RIPK3 and consequent necroptosis [128]. It is proposed that DAI might act as a sensor of a VV-induced pathogen-associated molecular pattern (PAMP). A lesser studied necroptosis inhibitor is the viral FLIP proteins (vFLIPs) identified in herpesviruses and the human poxvirus (molluscum contagiosum virus). Cellular FLIP proteins (c-FLIP) are key regulators of caspase-8 activity [129]. Among various c-FLIP multiple splice forms at the mRNA level, two major protein isoforms, namely c-FLIP long (c-FLIPL) and c-FLIP short (c-FLIPS) were studied extensively [121,130]. c-FLIPS inhibit apoptosis by blocking caspase-8 activation [121,131], whereas c-FLIPL: procaspase-8 heterodimers inhibit RIPK1-RIPK3-mediated programmed necrosis [132]. Hence, the mechanism by which vFLIPS inhibit necroptosis is still elusive.

#### DNA Viruses Mediated Anti-Necroptosis

The mechanism of interference of apoptosis and necroptosis by HSV was extensively studied among DNA viruses. Herpes viruses such as cytomegaloviruses, the alpha herpes viruses, HSV-1 and HSV-2, and EBV target the same protein, RIPK3, for inhibition of necroptosis. Some poxviruses such as BeAn 58,058 and Cotia poxviruses also possess proteins with high sequence similarity to the mammalian pseudokinase domain of MLKL and inhibit necroptosis by sequestering RIPK3 via prevention of MLKL phosphorylation in mouse and humans [133].

The first viral inhibitor of necroptosis or RIP activation (vIRA) identified was encoded by the M45 gene from MCMV [50,134]. The M45 gene encodes an inhibitor protein homologous to the ribonucleotide reductase (R1) at the C-terminus, however, with no enzyme activity and a RIP homotypic interaction motif (RHIM). M45 blocks necroptosis by disrupting RHIM-dependent interactions between RIP1 and RIP3 [50,134,135]. However, the action of M45 showed tissue-specificity with optimal viral replication in specific cell types in tissue culture [114,136]. M45 might inhibit ZBP1/RIPK3-induced necroptosis, as replication of the M45 RHIM mutant virus was restored in *Ripk3*^−/−^ or *Zbp1*^−/−^ cells and mice [134]. It is shown that M45 mutant viruses fail to infect mice [137,138] due to excessive activation of necroptosis [50,134]. In DAI/ZBP1-induced necroptosis, DAI/ZBP1senses viral RNA sensor [139] instead of DNA as previously thought [140]. It is shown that RIPK1 knockdown or inhibition fails to protect from MCMV-induced necroptosis, whereas inhibition of the viral transcriptional activator, immediate-early viral trans-activator (IE3) protected from MCMV-induced necroptosis [141]. Similarly, the human cytomegalic virus (HCMV) suppresses necroptosis without affecting the phosphorylation of RIPK3 and MLKL. Therefore, it is suggested that HCMV inhibition of necroptosis occurs at the stage following MLKL activation [142].

## 4. Cell Death Signaling in RNA Virus Infection

RNA viruses can induce cell death through multiple pathways through apoptosis, necroptosis, and possibly pyroptosis. RNA viruses such as influenza A virus (IAV), rubella virus, polyoma virus, zika virus (ZIKV) and Sendai viruses (SeV) release multiple copies of cellular contents into the extracellular environment, promoting inflammation and Type 2–mediated immune response [143] (Figure 3).

### 4.1. RNA Viruses Mediated Apoptosis

Apoptosis is one of the most common antiviral defense mechanisms. There are two pathways of apoptosis: intrinsic and extrinsic. Viral infection is one of the stimuli contributing to activating the extrinsic apoptosis pathway. The extrinsic pathway is regulated by membrane death receptors such as DR4/TRAIL-R1 and DR5/TRAIL-R2. The receptors TNFR1 and CD95 are activated by their ligands such as TRAIL, TNF, and FasL. The binding of ligands induces the activation of caspases, the main enzymes (proteases) involved in apoptosis [144]. Viruses have derived several mechanisms from modulating the proapoptotic signals of the host to prevent host cell death.

While large DNA viruses have the genetic makeup to synthesize accessory proteins to interfere with host cell apoptosis, lesser is known about RNA viruses exhibiting such an arrangement. One reason for this gap in understanding RNA virus response to apoptosis is that most RNA viruses replicate before cell death. However, few viruses are slow replicators, and certain RNA viruses have shown a simpler arrangement, such as the rubella virus inhibiting apoptosis via the anti-apoptotic activity of its capsid protein [145]. The capsid protein binds to Bax-inhibiting peptides (BIP), its pore-forming capacity as Bax forms hetero-oligomers upon binding with capsid protein [145,146].

It was reported that infection with ZIKV results in apoptosis of the neural progenitor cells (NPCs) resulting in ZIKV infection-related microcephaly. This occurs as a result of induction of conformational activation of Bax a mitochondrial pro-apoptotic protein by the ZIKV which subsequently results in the formation of Bax oligomers in the host cell mitochondria. This promotes an increase in the release of Cyt c, yet another pro-apoptotic factor, enhancing the loss of mitochondrial membrane potential and integrity, which results in cell apoptosis. The nonstructural protein 4B (NS4B) protein of ZIKV plays an important role in this, as it is involved in mitochondrial recruitment of Bax protein and induction of Bax conformational activation. This indicates that ZIKV infection directly impacts the intrinsic pathway of apoptosis by modulating the recruitment and activation of Bax [147]. Apart from this, infection with ZIKV also causes endoplasmic reticulum (ER) stress in the maternal placental trophoblast cells. This sustained ER stress results in the apoptosis of these trophoblastic cells by increasing the levels of both mRNA and protein of CHOP an inducer of apoptosis. Thus, the mechanism behind ZIKV-induced placental cell apoptosis is the activation of ER stress [148]. A recent study has also demonstrated that infection by nephropathogenic infectious bronchitis virus (NIBV) induces ER stress and apoptosis in the renal cells by activation of the GRP78/PERK/ATF-4 signaling pathway, which results in kidney damage. Activation of this pathway results in upregulation in the expression of pro-apoptotic proteins such as CHOP, Caspases, P53 and Bax whereas a downregulation in the expression of anti-apoptotic factors such as the Bcl-2 protein. This shows a relation between ER stress gene expression, apoptosis and renal injury during NIBV infection [149].

### 4.2. RNA Viruses Mediated Pyroptosis

Among the various inflammasomes, NLRP3, AIM2, IFI16, and retinoic acid-inducible gene-I (RIG-I) were associated with virus-infected cells. The NLRP3 is the most well-studied inflammasome and is activated by both DAMPs and viral components such as RNAs and pathogen-associated molecular patterns (PAMPs) [114]. The NLRP3 was subsequently identified to play a major role in influenza and SeV infection in macrophages [150]. In vitro studies have shown various viruses and viral PAMPs, including viral RNA analogs Poly (I:C) and ssRNA40 [150,151], transfected adenovirus DNA [152], and rotavirus (RV) double-stranded (ds) RNA, triggered the NLRP3 inflammasome [150].

The IFN-inducible protein AIM2 was recognized originally as a cytosolic DNA sensor [153] and binds directly to double-stranded viral or bacterial DNA, triggering inflammasome complex formation [108,154,155]. However, recent findings showed that AIM2 is also induced by RNA viruses [156]. The AIM2 contains a PYD domain and a C-terminal hematopoietic interferon-inducible nuclear proteins with a 200 amino acid repeat (HIN-200) domain. It interacts with ASC via its PYD domain to recruit caspase-1 forming the AIM2 inflammasome. The RNA viruses such as chikungunya virus (CHIKV), West Nile virus (WNV) and ZIKV were shown to invoke AIM2-mediated pyroptosis and secretion of IL-1β [157,158]. The RIG-I plays a crucial role in RNA virus recognition [159,160]. It consists of two N-terminal CARD domains, a C-terminal regulatory domain (CTD) and a central RNA helicase domain [161]. The CTD binds with dsRNA in a ubiquitin-chain-mediated process and undergoes conformational changes leading to its oligomerization and the subsequent interaction with RIG-1 and mitochondrial membrane antiviral protein (MAVS) via a CARD–CARD interaction. Thus, CARDs transmit the activation signal downstream. This results in the formation of MAVS dimer in mitochondria and leads to the expression of type I IFNs (IFN-α/β) and pro-inflammatory cytokines [160,162]. The RIG-I recognizes exclusively RNAs bearing blunt-ended, base-paired termini with 5′-diphosphate (5′pp)- and triphosphates (5′-ppp) [160,163]. Such a recognition criterion provides a strong self-non-self-distinction as host cytosolic RNAs (mRNA, tRNA, or rRNA) lack both 5′pp- and 5′ppp-termini [160,164]. The RIG-I mediated antiviral immunity was reported by cells infected with reovirus or transfected with the ds RNA segments of the reovirus genome [159,160].

During infection by influenza H7N9 virus, it activates pyroptosis mediated by gasdermin E (GSDME) in host lung alveolar epithelial cells. The cytosolic contents released as a result of pyroptosis trigger a cytokine storm. This indicates that GSDME activation is a key and unique mechanism for the pulmonary cytokine storm and lethal outcome of H7N9 virus infection [165]. During ZIKV infection, the expression of caspase-1 and GSDMD is also increased, which contributes to increased secretion of IL-1β and IL-18. This suggests that the NLRP3-mediated inflammasome activation may lead to pyroptosis in infected host macrophages thus resulting in their death. This indicates that in ZIKV-infected macrophages, apoptosis and pyroptosis occur concomitantly and may have potentially significant impacts on viral pathogenesis in humans [166]. Another study by He et al. demonstrated that caspase-1 and GSDMD-mediated pyroptosis by ZIKV and its effect on NPCs development ultimately led to microencephaly. The ZIKV exposure led to pyroptosis in neural cells, including astrocytes, NPCs and microglia, both in vivo and in vitro, experimentally linking the mechanism of ZIKV infection-causing microencephaly [167].

#### IFI16 Mediated Pyroptosis

The IFN gamma-inducible protein (IFI16), is a DNA sensor that evokes pyroptosis in human immunodeficiency virus (HIV) [168,169] and Kaposi sarcoma-associated herpesvirus (KSHV) infections [170]. IFI16-mediated pyroptosis plays an important role in reducing CD4^+^ T cells that are not productively infected with HIV [169,171,172]. IFI16 detects incomplete HIV reverse transcript accumulated in the cytosol, triggering an innate IFN-β response and inflammasome assembly that leads to caspase-1 activation. However, the mechanism of caspase-1 activation by IFI16 remains unclear [173]. Pyroptosis is not only considered protective in chronic infections such as HIV but it also results in a vicious pathogenic cycle in which dying CD4^+^ T cells release inflammatory signals that cause increased cell death [169]. Caspases-1 inhibitors are proposed as an ‘anti- acquired immunodeficiency syndrome’ (AIDS) therapy that targets the host instead of the virus [174]. Virus replication and dissemination are adversely affected by caspase-1-induced inflammation and pyroptosis. In turn, viruses have evolved proteins that target caspase-1 function and its downstream signaling, such as the PYD protein of a poxvirus [175] and influenza non-structural protein 1 (NS1) [176]. In addition, different pathogens have different mechanisms for interfering with inflammasome activity by encoding inhibitor proteins secreted to the cytosol [177].

Virus-induced pyroptosis is not only restricted to immune cells such as macrophages and monocytes but can also occur in epithelial cells. For instance, upon infection of differentiated human nasal epithelial cells (hNECs) and primary human nasal epithelial progenitor cells with human rhinovirus (HRVs), the epithelial NLRP3 inflammasomes mediated the hNEC pyroptosis via DDX33/DDX58–NLRP3–caspase-1–GSDMD axis. It was also observed that infection with HRVs induced secretion of lactate dehydrogenase (LDH), which is a typical property of pyroptosis. Thus, it was evident that the HRV-induced airway epithelial inflammatory response, i.e., secretion of IL-1β and pyroptosis, depends on the DDX33/DDX58–NLRP3–caspase-1–GSDMD axis, which was mediated by inflammasome formation [178]. Another study showed that infection of precancerous human bronchial epithelial cells PL16T with IAV-induced pyroptosis in the late stage of infection while apoptosis was observed in the early stages. In these cells, the type-1 IFN signaling pathway inhibited the apoptosis of PL16T cells via the expression of Bcl-xL anti-apoptotic genes. Thus, type-1 IFN stimulates pyroptosis in respiratory epithelial cells during IAV infection to initiate pro-inflammatory responses [179].

A study showed that mammalian adapted new Asian lineage avian influenza A (H7N9) virus infection in alveolar epithelial cells triggered cytokine storm by activating pyroptosis of these cells mediated by GSDME. Activated caspase-3 was found in the lung alveolar epithelial cells and also the GSDME levels increased as the infection progressed. It was thus evident that infection with H7N9 could activate and switch the cell death from caspase-3-mediated apoptosis to pyroptosis in alveolar epithelial cells expressing high levels of GSDME. Hence, it was proved that GSDME activation is a prime factor in host alveolar epithelial cell death post H7N9 infection [165]. Infection of intestinal epithelial cells with transmissible gastroenteritis virus (TGEV), a type of coronavirus, led to the production of pro-IL-1β, its processing and maturation via activation of caspase-1. Additionally, pyroptosis of these cells was observed via the production and cleavage of GSDMD and through the activation of NLRP3 inflammasome [180]. GSDME-mediated pyroptosis is also observed in enterovirus A71 (EV71) infection of HeLa and SK-N-SH cells. After infection with EV71, caspase-3 was shown to be activated and cleavage of GSDME at the amino acid pair of D270–E271 was noted. EV71 infection induced the release of LDH and induces pyroptosis in HeLa and SK-N-SH cells. Additionally, cells expressing GSDME showed a switch from EV71-induced apoptosis to pyroptosis. This suggests that GSDME cleavage mediated by caspase-3 played a crucial role in the pathogenesis of EV71 [181]. Another study reported that upon infection of intestinal epithelial cells with RV, the viral RNA helicase Dhx9, and inflammasome Nlrp9b recognized short stretches of dsRNA thus forming inflammasome complexes with the adaptor proteins such as ASC and caspase-1. This promoted IL-18 maturation and GSDMD-induced pyroptosis of RV-infected intestinal epithelial cells [182]. From these studies, it is evident that virus-induced pyroptosis can occur in epithelial cells of the host and is not only restricted to macrophages and monocytes.

### 4.3. RNA Viruses Induced Necroptosis

Some viruses are susceptible to necroptosis, including reovirus [183,184,185] while many, especially large DNA viruses of the class, poxviruses, herpesviruses, and IAV evolve different anti-necroptosis mechanisms. Necroptosis can be induced by various stimuli, including death ligands (TNFα, FasL, and TRAIL), IFNs, TLR ligands and microbial infection [125,186,187]. Based on the stimulants, necroptosis can be of three types: (A) Extrinsic necroptosis triggered by TNFα, (B) Intrinsic necroptosis by ROS, and (C) Ischemia mediated intrinsic necroptosis [45].

The mechanism of necroptosis is conventionally considered a fail-safe alternative to apoptosis, so whether it can restrict IAV or indeed any virus in the absence of apoptosis is still unclear. Shubina et al. demonstrated that mice lacking IAV-induced apoptosis, drive robust necroptosis-mediated antiviral immune responses and permit effective virus clearance from infected lungs. According to their findings, IAV-infected cells undergo both apoptosis and necroptosis. However, apoptosis is not a necessary component of host defense in antiviral environments, as necroptosis is independent, and it restricts the IAV as a stand-alone cell death mechanism that fully compensates for apoptosis’s absence [188].

TNFα-induced necroptosis is the most well-characterized of all necroptotic pathways. Necroptosis, in general, is centrally driven by the activation of the intracellular RIP family of serine/threonine kinases such as RIPK3 through interaction with an RHIM-containing protein in the following pathways: RIP1 (TNF family death receptors and type I IFNRs); TIR-domain-containing adapter-inducing interferon-β (TRIF) facilitates TLR3/4; DAI (M45/vIRA) mutant MCMV infection); HSV R1 (HSV infection in the mouse cells) [189]. The RIP family kinases are activated following the binding of TNF family cytokines such as TNFα, Fas and TRAIL to membrane receptors. Upstream targets include three RHIM-containing adapters, RIPK1, TRIF, and DAI/ZBP1/DLM1 [190,191] and MLKL [192]. RIPK3 integrates signals from upstream receptors and adapters. RIP1 forms a protein complex with RIP3, known as necrosome, via their RHIM domain [185,193]. Further, RIPK3 phosphorylates MLKL promoting oligomerization of MLKL [194,195]. MLKL then translocates to the plasma membrane, either promoting the pore formation in the plasma membrane by interacting with the amino-terminal of phosphatidyl inositol phosphate or facilitating the recruitment of Na^+^ or Ca^++^ ion channels in the plasma membrane [196,197].

Necroptosis mediates its signaling process through a series of proteins, including RIPK1/3 and MLKL, which regulate the inflammatory mode of cell death [46]. Virus-induced necroptotic cell death is caspase-independent. First, it activates RIPK1 and RIPK3 kinases resulting in the formation of the RHIM-dependent necrosome complex with pseudokinase MLKL. Further, subsequent activation of oligomerization of MLKL leads to membrane translocation and necrotic plasma membrane disruption [198]. Some RNA viruses, such as VSV or encephalomyocarditis virus (EMCV), promote inflammatory response by activation of the NLRP3 inflammasome through cytopathogenic effect-induced potassium efflux [199]. This molecular signaling mechanism activates cellular damage-associated molecular patterns, including high mobility group box protein 1 (HMGB1) [200]. Several reports highlighted that necroptosis is a “fail-safe” form of cell death, limiting viral spread while alerting the immune system to danger, but with apoptosis, many viruses have evolved strategies to limit necroptosis to promote viral replication. Mainly, it targets RIPK3 and other components of the necroptosis pathway to manipulate host defense and antiviral inflammation [186].

A recent study by Wen et al. demonstrated that ZIKV infection of astrocytes led to necroptotic cell death. In response to ZIKV infection, the release of pro-inflammatory cytokines, such as IL-6, IFN-β and IL-8 was observed. The ZIKV-infected astrocytes also showed an increased expression of phosphorylated RIPK1, RIPK3, and MLKL. Additionally, they observed upregulation of ZBP1 in ZIKV-infected astrocytes, which in turn activates RIPK3. Thus, according to their study results, RIPK3-dependent necroptosis proved to play an inhibitory role in ZIKV replication, indicating that necroptotic cell death could be beneficial to the host even though infected astrocytes died of the necroptotic signaling during the infection [201]. Another study has demonstrated that RV uses necroptosis along with apoptosis, promoting host cellular death during the late phase of infection. This is brought about by phosphorylation, which results in the activation of MLKL protein by RV. The phosphorylated form of MLKL oligomerizes and translocates to the plasma membrane of the cells infected by RV. This results in the loss of integrity of the host cell plasma membrane and the release of alarmin molecules such as HMGB1 which contribute to host cell death [202].

## 5. Severe Acute Respiratory Syndrome Coronavirus 2 (SARS-CoV-2) Infection Associated Cell Death Signaling

Coronavirus effectively infects new cells in the respiratory tract and lungs. In common, it enters our body through the nose or mouth, where it begins to disrupt the airways and increase the severity of pneumonia [203]. The infection mechanism begins with virus–host binding, where a viral spike protein (S1) attaches to host-specific angiotensin-converting enzyme 2 (ACE2) receptors on the cell surface to infect humans [204]. When the viral envelope fuses with the lipid membrane of host cells, the virus can release its genetic material into the interior of the healthy cell, causing it to become more contagious. Coronavirus contains RNA as its genetic blueprint, which serves as a chemical message, instructing the host cell machinery to read the template and translate it into proteins that form new viral particles [205]. Infected cells can develop and release millions of copies of the virus, subsequently triggering cell apoptosis, inflammatory responses, and cell death through caspase-8 activation in lung epithelial cells [206]. Activated caspase-8 propagates the apoptotic caspase signal, which has emerged as a master regulator of the three major cell death pathways, including apoptosis, pyroptosis, and necroptosis [104]. SARS-CoV-2 infections activate TNF-α and IFN-γ signaling pathways that trigger inflammatory cell death, leading to the production of multiple inflammatory cytokines [207] (Figure 4).

### 5.1. SARS-CoV-2 Mediated Apoptosis

Apoptosis is considered a silent mode of death mechanism. Apoptosis proves to be beneficial for the host after the invasion of viruses, as viruses cannot attach to the cells to cause disease. Recent reports demonstrate that the two accessory proteins of SARS-CoV-2 including ORF3a and ORF7b proteins, induce apoptosis through the TNF-α signaling pathway [208,209]. However, in severe coronavirus disease (COVID-19) patients, the result shows that induced T cell lymphopenia and redistribution of T cell populations in serum [210]. T cell triggers FasL production in large amounts [211], and it upregulates Bcl-2 family members, including Bax and Bak in COVID-19 [212]. Ren et al. demonstrated that SARS-CoV-2 triggers caspase-dependent apoptosis, which induces PDK1 kinase and inhibits the activation of PDK1-PKB/AKT signaling in Vero E6 cells (monkey kidney cells), HEK393T cells (human embryonic kidney cells), and HepG2 cells (hepatocellular carcinoma cells) [213]. SARS-CoV-2 infection activates caspase-8 and caspase-3-dependent inflammatory cytokine responses in the lung epithelial cells, and it causes severe lung damage in COVID-19 patients [206,214,215]. Therefore, the extrinsic apoptotic pathway is likely to be initiated due to the SARS-CoV-2 infection [216]. Inflammation is the initial reaction to this interaction, aided by pro-inflammatory signals. In several research findings, inflammatory cytokines such as IL-6, IL-2, IL-7, IL-8, IFN-γ, TNF-α, CXCL10, CCL5, GM-CSF, and IP-10 are induced by SARS-CoV-2-mediated apoptosis via receptors known as NLRs [206,207,217].

### 5.2. SARS-CoV-2 Mediated Necroptosis

Necroptosis is also known as “programmed necrosis”, a combination of necrosis and apoptosis. Viruses escape the apoptotic pathways majorly by blocking or mutating the caspases-1/8 with the help of cFLIPL and cFLIPS. Increasing evidence reports show that necroptosis plays a role when the apoptosis is compromised to battle the infection and ultimately protect the host. This generates a hyper-inflammatory response by cell lysis and releasing DAMPs and cytokines instead of the silent death of apoptosis [186,211]. The effectors of necroptosis including MLKL and RIPK3 are upregulated in the Calu-3 cells (lung epithelial cells) infected with SARS-CoV-2. The proposed model of SARS-CoV-2 infection induces inflammatory responses and cell death. Li et al. reported that SARS-CoV-2 infection induces cell death by activating caspase-8. Furthermore, the activation of caspase-8 promotes pro-IL-1β cleavage leading to the secretion of a mature IL-1β active p17 fragment (P17) through the necroptosis pathway [206].

### 5.3. SARS-CoV-2 Mediated Pyroptosis

Pyroptosis is another mechanism of cell death mediated mainly by the GSDMD, which is involved in the formation of pores in the membrane due to the high influx of sodium (Na^+^) and water, leading to swelling of the cell and rupturing [218]. GSDMD is cleaved by caspase-1, leading to the release of mature IL-1β and IL-18, inflammasome-dependent NLRP3 [211]. Human monocytes infected with SARS-CoV-2 trigger pyroptosis associated with caspase-1 activation, IL-1 β production, GSDMD-cleavage, and increased pro-inflammatory cytokines [219]. The LDH is another indicator of pyroptosis. When the membrane ruptures, it initiates a cascade by releasing viral particles into the circulation, attracting the cytokines and chemokines to generate a large inflammatory event [218]. Ferreira et al. have shown that IL-1β production was prevented by inhibiting NLRP3 (glyburide), caspase-1 (AC-YVAD-CMK) and enhanced the expression of IL-6 and TNF-α, suggesting that patients with severe COVID-19 infection display monocyte cell death, which is caspase-1-dependent, and exhibit cell lysis with high levels of IL-1β [219].

## 6. Lesser-Known Non-Apoptotic RCD Associated with Viral Infection

### 6.1. Ferroptosis in Viral Infection

Ferroptosis is a type of non-apoptotic regulated and autonomous cell death pathway that is largely dependent on iron (Fe)-mediated formation and accumulation of free lipid radicals [24,220]. It results from oxidative alterations of the cellular microenvironment marked by the lipid peroxide radicals formed due to Fe ions. Further, the Fenton reactions and the inability of the glutathione–glutathione peroxidase 4 (GSH–GPX4) antioxidant system to remove the free radicals results in the accumulation of lipid radical ions giving rise to cell lipotoxicity eventually resulting in cell death referred to as ferroptosis [24]. In ferroptosis, Fe enters the cell as a ferrous (Fe^2+^) ion via the transferrin receptor protein 1 (TfR1). While inside the cell, Fe^2+^ is oxidized to ferric (Fe^3+^) ions. These ions are then enclosed in the endosomes with an acidic environment, where they are again reduced to Fe^2+^. Further, the Fe^2+^ is then transported into the cell cytoplasm by a transporter called divalent metal transporter 1 (DMT1). The Fe^2+^ present in the cytoplasm then binds to ferritin, which in ferroptosis is degraded by lysosomes in a process known as ferritinophagy, which releases free Fe into the cytosolic pool of labile Fe, promoting its accumulation and thus inducing ferroptosis [221]. Apart from this, non-ferritin Fe reacts with O_2_ molecule, e.g., such as H_2_O_2_ and the hydroxyl (OH) radicals generated in the Fenton reaction, which reacts with polyunsaturated fatty acids (PUFAs) which forms lipid peroxide radicals [222]. Under normal conditions, the activity of these lipid peroxide radicals is kept under check by the GSH–GPX4 anti-oxidant system which reduces the lipotoxicity. However, the inhibition of this antioxidant activity results in loss of control over lipid peroxidation which in turn facilitates ferroptosis. Apart from this, the increased Fe uptake or/and inhibition of Fe export promotes ferroptosis by increasing the free Fe content in the cell [12].

Ferroptosis is different with respect to its features from other forms of cell death. It is marked by characteristic features such as small mitochondria with increased membrane density, reduction or absence of mitochondrial cristae, and ruptured outer membranes, electron-lucent cell nucleus due to damage by lipid peroxide radicals [220,223,224]. With respect to biochemical features, ferroptosis is characterized by a reduction in GSH and reduced activity of GPX4 along with lipotoxicity [224]. Recent findings suggest that viruses have evolved mechanisms that modulate and exploit ferroptosis in host cells. As Fe is an important metal element for different cellular enzymes involved in the maintaining of host physiology as well as inefficient viral replication, it acts as a source of competition, in terms of nutrition, between the host and the virus. In this view, viruses have evolved strategies to interfere with the host cell ferroptosis for their own benefits, such as an interruption of host cell Fe uptake and the suppression of the host antioxidant response system. Apart from this, some viruses can also utilize host Fe transporters as viral receptors to enter the host cell thus promoting viral infections by promoting ferroptosis [12].

Excessive cellular Fe concentration is the primary cause of ferroptosis; however, virus replication requires Fe as an important raw material. Thus, during infections by viruses such as HIV, HCMV, VV, HSV1, and Hepatitis B virus (HBV), more Fe is transported into the host cell. This triggers the Fenton reaction and generates large amounts of lipid peroxide radicals. These lipid radicals cannot be eliminated by the reduced amount of GPX4 ultimately resulting in ferroptosis [225]. For instance, neuraminidase (NA) of the influenza virus results in the degradation and deglycosylation of the lysosomal-associated membrane protein (LAMP) of the lysosomes of the infected cells. This causes digestion and rupture of the host lysosomal membrane resulting in the release of a high concentration of Fe present in the lysosome into the cytosol thereby causing a cytosolic accumulation of labile Fe. The accumulated Fe then mediates Fenton lipid peroxidation, which further leads to cell death due to lipid peroxide accumulation [226,227,228,229]. In HCV infection, hepcidin, an Fe homeostasis protein, is downregulated. This causes an increase in ferritin levels and saturation of transferrin, which contributes as a crucial factor for Fe accumulation in the liver cells during HCV infection resulting in the progression of the infection and resistance to treatment [230,231,232]. Similarly, HIV is associated with the reduction of GSH by promoting ROS production (oxidative stress) in plasma and lung epithelial lining fluid which results in an increase in the concentration of lipid peroxidation products such as malondialdehyde (MDA) in the serum of HIV patients [233]. Apart from this, the levels of GPX-4 were also found to be reduced in HIV infections resulting in ferroptosis of the host cell [234].

Ferroptosis also plays an important role in SARS-CoV-2 pathogenesis. Since, a decrease in levels of GSH, GPX4 inactivation, perturbed metabolism of Fe, and upregulation of PUFA peroxidation are key features of COVID-19 disease, SARS-CoV-2 may trigger ferroptosis in the cells of different host organs, which contributes to multiorgan damage [235]. During SARS-CoV-2 infection, the expression of GPX4 is reduced. In the absence of GPX4, GSH cannot be peroxidated and hence cannot reduce lipid ROS generated by the Fenton reaction. This results in lipid ROS accumulation causing lipid peroxidation and ultimately ferroptosis [236]. Previous studies have also reported that ferroptosis may also play a potential role in the development of COVID-19-related brain injuries as Fe is one of the most abundant metals found in the brain and is needed for normal physiological processes of the brain. The brain is also sensitive to oxidative stress generated by ROS and lipid peroxidation due to its high level of PUFAs as the cell membranes of the neurons contain high amounts of PUFAs, thus being susceptible to ROS-mediated oxidation caused by a cytokine storm, a key feature of SARS-CoV-2 infection in COVID-19 [237,238]. Ferroptosis may also occur due to the dysregulation of Fe homeostasis in COVID-19 patients. Additionally, upon infection with SARS-CoV-2, IL-6 present in the cytokine storm increases the intracellular levels of ferritin and the production of hepcidin. This leads to a decreased Fe efflux from the cells. The intracellular ferritin is then degraded by a process, as previously described, referred to as ferritinophagy. The ferritinophagy facilitates ferroptosis through the degradation of ferritin and releasing high amounts of labile Fe in the cell [236]. Thus, ferroptosis is an important mechanism involved in multiple organ failure in COVID-19 and might serve as a new treatment target.

This suggests the possible occurrence of ferroptosis as a cell death mechanism along with dysregulated cell metabolism during viral infections. However, the exact underlying mechanism that viruses use to interrupt the Fe metabolism remains elusive thus necessitating its further study. The decipherment of the regulation involved in ferroptosis and its role in viral infection is still critical and can help in the development of new antiviral therapeutic avenues.

### 6.2. Entosis in Viral Infections

Entotic cell death is a type of cannibalism in which one cell is engulfed and killed by another cell of the same type with the use of cell adhesion molecules, cytoskeleton and energy expenditure. Entosis commonly occurs in epithelial cells and is influenced by cell detachment from the basement membrane. A characteristic feature of entosis is the formation of cell-in-cell (CIC) structures [239]. Once the cells are engulfed, they can either be eliminated by regulated cell death mechanisms inside an endosome, the entotic vacuole, via an autophagy-related process referred to as LC3-associated phagocytosis (LAP) [240]. Apart from this, the engulfed entotic cell can also either divide and proliferate within the host cell engulfing it or can escape without any degradation [241,242]. During entosis, the inner entotic cell that gets engulfed by the other cell enters the host cell by activation of Rho proteins. This is followed by adhesion bond formation mediated by proteins such as β-catenin, E-cadherin and calcium (Ca^2+^) ions and through actin and myosin cytoskeletal filaments [243]. The inner cell is then surrounded by a double-membraned entotic vacuole, which is a characteristic of entosis [242]. This causes either the death of the inner cell, or that of the outer cell, or death of both cells, and survival of both cells.

Previous findings have reported the role of entosis in viral infections such as those caused by EBV or HIV. Entosis in this case is reported as an in-cell infection. This in-cell infection is considered an important mechanism involved in the spread of EBV from infected B cells to the surrounding epithelial cells, thus, predisposing the host to EBV-associated cancers [244,245]. In this, B cells act as the carriers of the virus to transfer the infection into the epithelial cells by utilizing cell adhesion protein-dependent conjugation. Entosis in EBV infection occurs by the ingestion of the whole EBV-infected B cells by the epithelial cells resulting in the formation of CIC structures [245]. The epithelial cells infected by this mechanism then start expressing viral gene products and produce virions upon stimulation, which then infect either naïve B cells or epithelial cells altered host cell tropism of EBV [244]. However, entosis is not unique to EBV and was also reported to be a potential mechanism for the effective transmission of HIV between susceptible CD4^+^ T-helper cells [245]. Direct cell–cell transfer of the virus is an efficient mechanism of viral dissemination within an infected host, exploited by human HIV-1. This direct transmission of the virus occurs via the host cell receptor engagement by HIV envelope (Env) protein. Further, it directs the recruitment of HIV receptors in an actin-dependent manner along with cell adhesion molecules to the interface. It forms a stable adhesive junction across which HIV is transferred from the infected cell to the other host target cell [246]. This suggests a potential role of ferroptosis in cell death during viral infections. However, the exact underlying mechanism that viruses use to modulate ferroptosis remains elusive thus necessitating its further study, especially with the emergence of viral pandemics such as COVID-19.

### 6.3. Methuosis in Viral Infection

Methuosis is a recently added type of non-apoptotic cell death (caspase-independent). It is derived from the Greek word ‘methuo’ (meaning to drink to intoxication) as it is characterized by the accumulation of large fluid-filled vacuoles in the cytoplasm originating from macropinosomes. Micropinosomes are formed by localized signaling mechanisms that are associated with cell surface ruffles, involving phospholipids and enzymes which regulate the actin cytoskeleton [247]. Methuosis was reported for the first time by Chi et al. who highlighted the ectopic expression of constitutively active oncoprotein, H-Ras (G12V) [248]. Additionally, it was reported that instead of cell proliferation, gastric carcinoma and glioblastoma cells showed accumulation of many large, phase-lucent vacuoles in the cytoplasm, which over a period of time underwent caspase-independent cell death. Based on this, it was proposed that endosomal vacuolization results from constitutive activation of Ras oncoprotein, not only because of the increase in macropinocytosis but also because of a disruption in the lysosomal fusion of macropinosomes and normal endocytic trafficking [247]. The key downstream molecule of methuosis is Rac1, a member of the Rho family of GTPases. It plays an elemental role in the initial stages of methuosis, i.e., macropinosome formation and trafficking. An increase in expression of ectopic Ras beyond the levels required for activation of the canonical growth-stimulatory phosphatidylinositol 3-kinase (PI3-K) and growth factor-regulated extracellular signal-related kinase (ERK) pathways. The elevated levels of endogenous activated Rac1 are reported to lead to the formation of cytoplasmic vacuoles. Another small GTPase ADP-ribosylation factor 6 (Arf6), involved in macropinosome recycling, is also shown to be affected by constitutive Ras expression and is involved in vacuolization observed during methuosis [249]. Another observation is that as Rac1 GTPase levels increased, there was a decline in levels of Arf-6 GTPase. This was regulated by Rac-1-mediated activation of GIT-1 which controls levels of Arf6. Thus, it is proposed that Ras-induced vacuolization is a consequence of the combined effect of an increase in the formation of macropinosomes and a decrease in the recycling of macropinosomes. Another mechanism of methuosis is the one caused by synthetic small molecules such as indole-based chalcones such as 3-(2-methyl-1H indol-3-yl)-1-(4-pyridinyl)-2-propen-1-one (MIPP). The MIPP can lead to the accumulation of cytoplasmic vacuoles at even low concentrations [250]. The morphological characteristics of these cells dying by indole-based chalcones-induced methuosis were similar to methuosis induced by Ras and thus supported a non-apoptotic cell death mechanism. Methuosis induced by such molecules is a result of their interaction with specific protein targets, mainly the structural and regulatory components of the early or late-stage pathways of endocytosis.

Compared to the autophagosomes seen in autophagy, these Ras-induced vesicles observed in methuosis have different morphology. They are phase- and electron-lucent and have a single membrane as opposed to the double membranes of autophagosomes [251]. Typical features of apoptosis such as chromatin condensation, cell shrinkage, and plasma-membrane blebbing are not seen in methuosis and activation of caspases is not a pre-requisite [251,252]. These vacuoles are characterized by the presence of extracellular fluid and have a neutral pH [253]. Methuosis is an important mechanism employed by certain viruses during pathogenesis. For instance, SV40 infection of African green monkey cells showed the presence of cytoplasmic vacuoles, typical of methuosis during the late stage of infection. The formation of vacuoles was found to be triggered by the interaction between GM1 and VP1 at the cell surface. Additionally, activation of the Ras-Rac1-MKK4-JNK signaling pathway was observed during the late stages of SV40 infection which resulted in the formation of vacuoles thus facilitating cell lysis and release of progeny viruses [254].

Methuosis is a relatively new form of cell death mechanism being discovered and hence not many studies are available on methuosis as a mechanism in viral pathogenesis. Apart from the other viral diseases, SARS-CoV-2, a novel coronavirus has caused a worldwide pandemic of the human respiratory illness, COVID-19, and hence more research on cell death mechanisms such as parthanatos as a mechanism used by SARS-CoV-2 for its survival and propagation in the host needs to be carried out.

### 6.4. Parthanatos in Viral Infection

Parthanatos is a newly discovered cell death mechanism. It occurs as a result of the overactivation of the nuclear enzyme poly (ADP-ribose) synthetase 1 or poly (ADP-ribose) transferase 1, (PARP-1). The term ‘Parthanatos’ is derived from ‘par’ (for PAR polymer, synthesized following PARP-1 activation), and ‘Thanatos,’ meaning death in Greek mythology [255,256,257]. Parthanatos is independent of caspases, but it highly depends on the nuclear translocation of the mitochondrial-associated apoptosis-inducing factor (AIF). The translocation of this mitochondrial protein AIF to the nucleus, sometimes by poly (ADP-ribose) (PAR) generated because of PARP overactivation, leads to large-scale chromatin condensation and DNA fragmentation and in turn cell death [258]. Under normal physiological conditions, the enzyme PARP-1 regulates cellular homeostasis and preserves genomic stability [259,260,261]. It is also involved in cell differentiation, malignant transformation, cell division, DNA replication, mitochondrial function and cell death [260,262,263].

Parthanatos is quite different from other mechanisms of cell death such as apoptosis and necrosis and hence, a deeper understanding of the molecular mechanisms underlying the same is necessary. Initially, it was assumed that excessive activation of PARP-1 led to massive depletion of cellular NAD^+^ and as a result, caused cells to commit suicide [264,265]. NAD^+^ is essential for the synthesis of PAR and thus ATP is required for the synthesis of NAD^+^. Thus, according to the suicide hypothesis, there is a massive decrease in levels of cytosolic and nuclear NAD^+^ post overactivation of PARP-1. In spite of this, to date, there is no direct and convincing evidence supporting the role of energy depletion as a main cause of parthanatos. It was also observed that in cells lacking poly (ADP-ribose) glycohydrolase (PARG), activation of PARP-1 led to cell death via parthanatos in the absence of NAD^+^ depletion [266]. The PAR polymer generated upon overactivation of PARP-1 is a key signaling molecule involved in the parthanatos cascade [267]. Upon stimulation of cells with stress such as N-methyl-D aspartate (NMDA) excitotoxicity, nitrosative or oxidative stress and DNA alkylating agents, there is PARP-1 overactivation along with PAR levels. This PAR migrates to the cytosol, from where it signals mitochondrially localized flavoprotein AIF, to undergo nuclear translocation. This results in large-scale DNA fragmentation, chromatin condensation and eventually, cell death [268]. The AIF is a mitochondrial protein, its nuclear translocation is an important step that links parthanatos with the mitochondria [267]. The pool of AIF in the outer, as well as inner mitochondrial membrane, is essential for parthanatos. Until now, the exact role of AIF in DNA fragmentation is unknown; however, few studies suggest that AIF may recruit endogenous proteases or nucleases to cause fragmentation or, interact with DNA and increase the DNA vulnerability to these molecules. For instance, it is observed that cyclophilin A interacts with AIF, leading to the formation of a pro-apoptotic complex of DNA-degradation, which may act as a co-factor for nuclear translocation of AIF and AIF-dependent chromatolysis during cerebral hypoxia-ischemia conditions [269,270]. However, the exact role of cyclophilin A and AIF in the formation of the pro-apoptotic complex is still unknown. Another molecule interacting with AIF is mammalian endonuclease G (endoG), which is shown to interact with CPS-6, leading to DNA degradation. However, it does not cause cell death following parthanatos in mammals which suggests that there is a parthanatos AIF-associated nuclease (PAAN) that has not been identified in vertebrates to date [271,272].

Several reports have shown that parthanatos is an important mechanism employed by certain viruses for their survival and propagation. Retinal tissue destruction is a characteristic of AIDS-related HCMV retinitis infection. Parthanatos-associated mRNAs and proteins were found in mice with retrovirus-induced immunosuppression during the onset of retinal necrosis, which occurred ten days after MCMV infection (MAIDS). The immunocytes infected with HCMV exhibited unique parthanatos-like features such as activation of PARP-1 (caspase-independent) and extensive DNA fragmentation [13]. Studies have also shown that cells infected with herpes simplex virus encephalitis (HSVE) and cytomegalovirus encephalitis (CMVE), underwent parthanatos by virus-induced PARP and translocation of AIF [273].

Given that parthanatos is a relatively newer mechanism of cell death being identified, fewer studies have reported its involvement in viral pathogenesis and propagation. Apart from the other viral diseases, SARS-CoV-2 is a novel coronavirus that has caused a worldwide pandemic of the human respiratory illness, COVID-19, and hence more research on cell death mechanisms such as parthanatos used by SARS-CoV-2 for its survival and propagation needs to be carried out.

## 7. Conclusions and Future Perspective

This review highlights multiple cell death response pathways in viral infection. Currently, considerable effort is needed to determine the pro-inflammatory mediated molecular mechanisms of PANoptosis (apoptosis, necroptosis and pyroptosis) in cell death. During viral infections, cell death mechanisms are used to destroy infected cells and restrict viral replication. In particular, an increased apoptotic response can compromise epithelial integrity, resulting in barrier disruption and adverse results. Targeting highly pathogenic virus-induced caspases implicated in virus infection and disease severity. Although targeting caspases and related pathways may be a promising intervention, caspase signaling may still be paramount for functional and balanced immune activity against viral infection. On the other hand, other mediators such as TNFRs, the Bcl-2 family, Cyt c, Apaf-1 and IAPs, also play a major role in cell death signaling and impact pathology. Exploring the relationship between disease-specific and cell death can provide a theoretical basis, especially focusing on distinguishing between physiological and pathological PCD (Table 1). Thus, targeting cell death-specific pathways study may be a promising avenue for new host-directed therapy to combat viral infections.

Molecular mechanisms underlying apoptosis inhibition or activation need to be better understood. Research into the mechanisms, cellular receptors, and/or microbial factors involved in the modulation of apoptosis may provide insights into the host–pathogen relationship and provide new therapeutic targets. Several techniques are available for measuring specific characteristics associated with cell death mechanisms. Rather than stating experimental results in terms of an apoptosis rate or cell death percentage, experimental results should be explained in terms of the methods used. Such a report will be able to better understand the type of cell death being measured. By studying the molecular mechanisms leading to cellular death and the outcomes of cellular death related to inflammation and immune responses, we will be better able to identify novel pathways of cellular dysfunction and achieve a deeper understanding of many pathological conditions that affect human health. In order to develop innovative therapies for viral infections, there is a need to understand how host responses are modulated and how their signaling mechanisms work. Over the last few years, proximity labeling approaches have been adapted to identify weak and transient protein–protein interactions in addition to co-immunoprecipitation methods. Biotinylation of proximate proteins in living cells is one of the techniques utilizing modified or chimeric soybean peroxidases (APEX). Combining molecular level labeling and mass spectrometry-based studies may help understand protein–protein interaction and the diversity of processes mediating cell death.

## Figures and Tables

**Figure 1 ijms-23-07023-f001:**
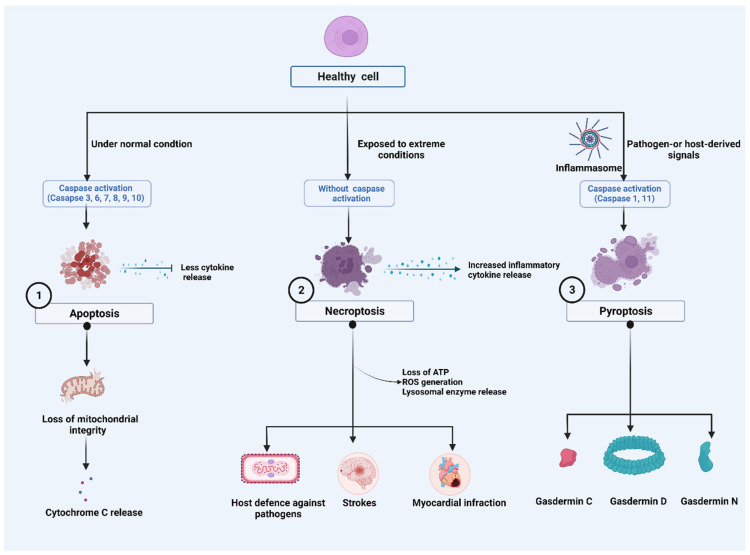
Major cell death pathways: (**1**). Harmless apoptosis: Mitochondria play a central role in the initiation of apoptosis. During early phases of cell death, cytochrome c (Cyt c) is often released from mitochondria. (**2**). A pathogenic mediator, necroptosis, plays a role in a variety of diseases and in the fight against viral infection. Despite having caspase activation, necroptosis plays a role in triggering and amplifying inflammation. Inflammatory responses are amplified by the activity of cytokines, which activate pro-inflammatory genes and lead to regulated cell death. Stress, infection, and injury can initiate the response by inducing regulated cell death directly as well as activating immune cells and cytokines. Receptor interacting protein-mediated reactive oxygen species (ROS) production, loss of ATP and lysosomal release can reactivate necroptosis, forming a positive feedback loop mechanism. (**3**). Orchestration of NLR family pyrin domain containing 3 (NLRP3) Inflammasome: Pathogen-or host-derived signals induce the activation of inflammatory caspases, such as caspase-1/4/5/11. These caspases cleave gasdermin D (GSDMD) resulting in the separation of the N- and C-terminal domains (GSDMD Nterm, GSDMD Cterm). GSDMD membrane pore formation constitutes the mechanism of pyroptotic cell death.

**Figure 2 ijms-23-07023-f002:**
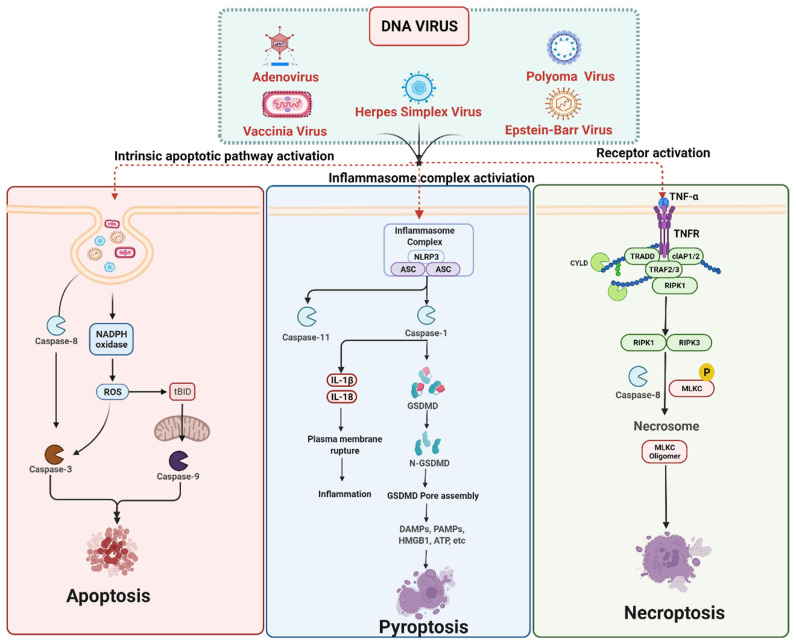
Cell death signaling mechanism by DNA virus infection: Mechanisms responsible for apoptosis, necroptosis, and pyroptosis, caused by RNA viral infections. Intrinsic apoptotic pathway mediates its signaling through NADPH oxidase (nicotinamide adenine dinucleotide phosphate oxidase), reactive oxygen species (ROS), caspases-3/8/9, Bcl-2-family protein truncated BID (tBID) promotes mitochondrial outer membrane permeabilization (MOMP) in apoptosis. In pyroptosis, inflammasome complex contains NLR family pyrin domain-containing 3 (NLRP3), apoptosis-associated speck-like protein containing a CARD (ASC), interleukin 1β (IL-1β), interleukin 18 (IL-18), gasdermin family (GSDMD). In necroptosis, inflammatory cytokines activate their signal through tumor necrosis factor (TNF), tumor necrosis factor receptor (TNFR), TNFR-1 associated death domain protein (TRADD), TNF receptor-associated factor 2 (TRAF2), receptor-interacting serine/threonine-protein kinase 1 (RIPK1), receptor-interacting serine/threonine-protein kinase 3 (RIPK3), phosphorylated mixed lineage kinase domain-like (p-MLKL), high mobility group box protein 1 (HMGB1), adenosine triphosphate (ATP), damage-associated molecular patterns (DAMPS), pathogen-associated molecular patterns (PAMPS), hepatitis C virus (HCV), Epstein–Barr virus (EBV), adenovirus, herpes simplex virus 1 (HSV-1), polyoma virus.

**Figure 3 ijms-23-07023-f003:**
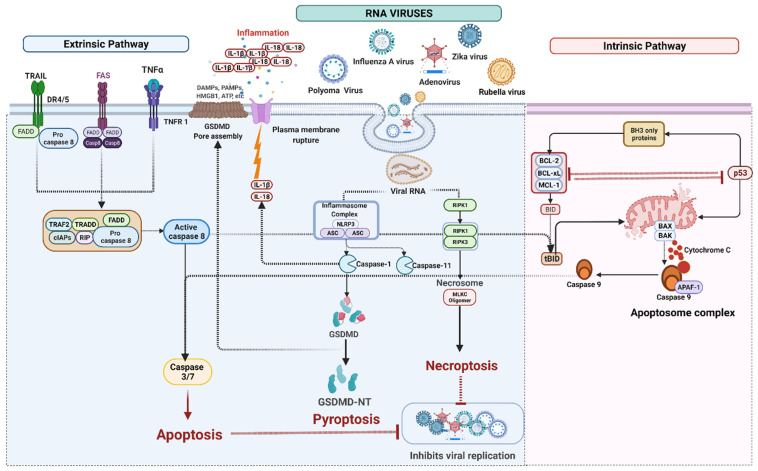
Cell death signaling mechanism by RNA virus infection: Mechanisms responsible for apoptosis, necroptosis, and pyroptosis, caused by RNA viral infections. RNA virus-induced cell death pathways. In extrinsic apoptosis, death receptors (DRs) 4 and 5 trigger caspase-8 activation by external ligands such as TNF-related apoptosis-inducing ligand (TRAIL) and Fas-associated via death domain complex [(tumor necrosis factor receptor type 1-associated death domain protein (TRADD), Fas-associated protein with death domain (FADD), TNF receptor-associated factor 2 (TRAF2). Z-DNA-binding protein 1 (ZBP1; also called DAI)] detects viral RNA and forms a FADD-caspase-8 death-promoting complex with receptor-interacting protein kinase (RIPK) 1 and 3 via RIP homotypic interaction motif (RHIM) interactions. The activation of caspase-8, the initiator caspase, initiates the cleavage and activation of effector caspases (caspase-3/7), which results in cell death. When caspase-8 levels are low or chemically inhibited during necroptosis, RIPK1 and RIPK3 are also activated. RIPK3 activates and phosphorylates mixed lineage kinase domain-like (p-MLKL), which oligomerizes and inserts into the plasma membrane, forming a pore that releases damage-associated molecules and interrupts cellular functions. Mitochondrial apoptosis is caused by cellular stressors, such as RNA viruses, which form the Bcl-2 associated X, apoptosis regulators (Bax/Bak) pores. These defects result in the release of cytochrome c (Cyt c) from mitochondria into the cytosol. Cyt c release is responsible for the assembly of the Cyt c. An apoptosome complex composed of caspase-9, Cyt c, and apoptotic protease activating factor 1 (Apaf-1) cleaves caspase-3/7 to induce apoptosis. The NLR family pyrin domain-containing 3 (NLRP3) inflammasome is triggered by viral pathogen-associated molecular patterns (PAMPs), in which caspase-1 cleaves gasdermin D (GSDMD), in order to release the N-terminal domain of GSDMD (GSDMD-NT). GSDMD-NT forms a membrane pore that facilitates the release of interferon-gamma (IFN-γ), interleukins- (IL-1β and IL-18) and damage-associated molecular patterns (DAMPs) such as high mobility group box 1 (HMGB-1). In addition, with an understanding of new activities for apoptotic caspase-3, it is presently unclear if caspase-3 cleaves GSDMD during an RNA viral infection to inhibit its pore-forming ability and/or if caspase-3 cleaves gasdermin E (GSDME) during apoptosis to release its N-terminal domain and trigger pyroptosis. zika virus (ZIKA), adenovirus, human polyomavirus (HPyV), rubella virus (rubivirus), and influenza A virus (IAV).

**Figure 4 ijms-23-07023-f004:**
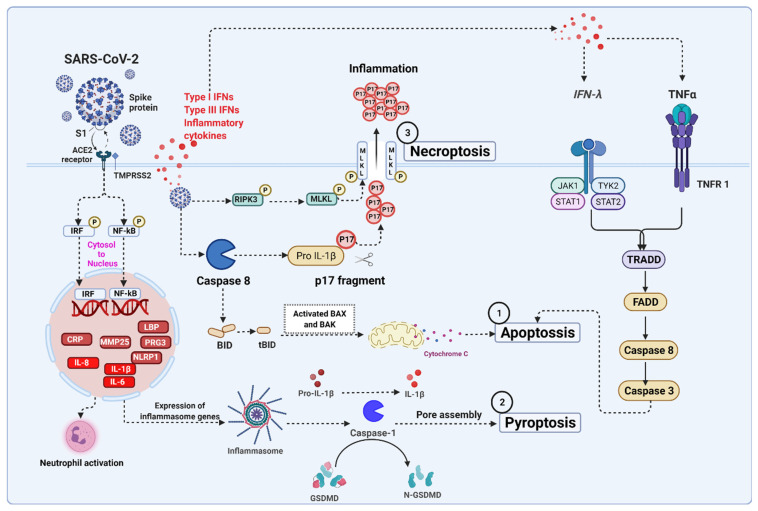
Cell death signaling in severe acute respiratory syndrome coronavirus 2 (SARS-CoV-2) infection: Mechanisms of cell death induced by proteins and cytokines produced by SARS-CoV-2. An increase in inflammatory cytokines is caused by the stimulation of nuclear factor kappa B (NF-κB) signaling by SARS-CoV-2. In response to binding of the angiotensin-converting enzyme 2 (ACE2) receptor, tumor necrosis factor-alpha (TNF-α) and interferon-gamma (IFN-γ) have a multiplicity of effects: (1) apoptosis is induced by the intrinsic pathway; (2) necroptosis is triggered by the phosphorylation of mixed lineage kinase domain-like (MLKL); (3) (A) pyroptosis occurs when inactive gasdermin is cleaved to form membrane pores by its active form. (B) The envelope (E) protein facilitates Ca^2+^ leakage from endoplasmic reticulum into the cytosol, and SARS-CoV-2 mediates K^+^ efflux from the viral membrane to the cytosol. This causes mitochondrial damage as well as reactive oxygen species (ROS) production, thereby triggering pyroptosis induced by NLR family pyrin domain-containing 3 (NLRP3) inflammasome. (C) SARS-CoV-2 enhances the pyroptosis pathway by promoting TNFR-associated factor 3 (TRAF3) -mediated ubiquitination of apoptosis-associated speck-like protein containing a CARD (ASC). (D) During infection with SARS-CoV-2, crosstalk between main cell death pathways causes PANoptosis (pyroptosis, apoptosis and necroptosis).

**Table 1 ijms-23-07023-t001:** Summary of viral infection mediated known and lesser-known RCD (regulated cell death) mechanisms.

Type	Morphological Features	Mechanism of Action	Major Regulators	References
Apoptosis	Cell shrinkage; fragmentation into membrane-bound apoptotic bodies and phagocytosis	- EBV infection inhibits the translation of the apoptosis inhibitor BRUCE and promotes apoptosis - HSV encodes caspase inhibitors, such as vICA that inhibit CD95 death-inducing signaling complex-mediated apoptosis thus, protecting cells from TNF and FasL-induced apoptosis. - ZIKV induces conformational activation of Bax pro-apoptotic protein which subsequently results in the formation of Bax oligomers in the host cell mitochondria promoting an increase in the release of cyt c enhancing the loss of mitochondrial membrane potential and integrity	- Caspases - Cytochrome c (Cyt c) - Apoptosis-protease activating factor 1 (Apaf-1) - Fas ligand and - TRAIL (tumor necrosis factor-related apoptosis-inducing ligand) - Death receptor 5 (DR5)	[29,97,98,101,147]
Necroptosis	Swelling of organelles, cell collapse, loss of membrane integrity and nuclear chromatin deficiency	- VSV or EMCV promote inflammation by activating NLRP3 inflammasome through cytopathogenic effect-induced potassium efflux, thus activating cellular damage-associated molecular Patterns such as HMGB1 - The vaccinia virus E3L gene encodes for an innate immune evasion protein, E3, with the N-terminus Z-form nucleic acid binding (Zα) domain and competes with DAI to prevent DAI-dependent activation of RIPK3 and consequent necroptosis	- Receptor-interacting protein kinase 3 (RIPK 3) - Mixed lineage kinase domain-like pseudokinase (MLKL)	[44,128,200,274]
Pyroptosis	DNA fragmentation, Condensation of nucleus, pore formation on plasma membrane, cellular swelling and subsequent rupturing of cells	- HSV-1 induces gasdermin -dependent pyroptosis through NLRP3 inflammasomes, inducing the protection of IL-1β and activating caspase-1 release. It can also activate the non-canonical inflammasome without NLRP3, ASC, and caspase-1 activation, which results in the activation of IL-1β but not IL-18. - CHIKV, WNV and zika virus were shown to invoke AIM2-mediated pyroptosis and secretion of IL-1β	- Caspase-1 - Gasdermin (GSDM)	[24,119,157,158,275]
Ferroptosis	Mitochondrial condensation, disappearance of cristae followed by outer mitochondrial membrane rupturing	- SARS-CoV-2 triggers ferroptosis by GSH–GPX4 inactivation in the host cells of different organs, which contributes to multi-organ damage - HIV reduces GSH by promoting ROS production resulting in an increase in the concentration of lipid peroxidation products (MDA). Apart from this, the levels of GPX-4 have also been found to be reduced in HIV infections resulting in ferroptosis of the host cell	- Glutathione–glutathione peroxidase 4 (GSH-GPX4) - Divalent metal transporter 1 (DMT1)	[233,234,236,276]
Entosis	Cell-in-cell structure in which viable cells invaded other cells	- Promotes transmission of HIV between CD4+ T-helper cells via the host cell receptor engagement by HIV Env protein which directs the recruitment of HIV receptors in an actin-dependent manner along with cell adhesion molecules to the interface, which forms a stable adhesive junction across which HIV is transferred from the infected cell to the other target cell	- RhoA-GTPase - RhoA effector kinases Rho-kinases I and II (ROCK I/II)	[14,246]
Methuosis	Accumulation of large fluid-filled vacuoles in the cytoplasm	- SV40 triggers the interaction between GM1 and VP1 at the cell surface along with activation of Ras-Rac1-MKK4-JNK signaling pathway, leading to cytoplasmic vacuole formation	- Rac1 GTPase	[249]
Parthanatos	Condensed and shrunken nuclei, disintegration of membranes and cells show positivity for propidium iodide within a few hours	- HCMV promotes parthanatos of immune cells by activation of PARP-1 (caspase-independent) and extensive DNA fragmentation	- Poly (ADP-ribose) transferase-1 (PARP-1) - Mitochondrial-associated apoptosis-inducing factor (AIF).	[13,256,257,258]

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
