# Peer review of "Revisiting Regulated Cell Death Responses in Viral Infections"

_ijms, 2022, doi:10.3390/ijms23137023_

Round 1
Reviewer 1 Report
Manuscript ID: ijms-1780539
Title: Revisiting Regulated Cell Death Responses in Viral Infection
Authors: Devasahayam Arokia Balaya Rex et al.
The manuscript is a very interesting and well-written review article presenting the current knowledge on regulated cell death responses in viral infection. Knowledge presented by the authors is necessary and useful for basic scientists as well clinicians. Molecular mechanisms underlying apoptosis inhibition or activation needs to be better understood. Research into the mechanisms, cellular, receptors, and/or microbial factors involved in the modulation of apoptosis, necroptosis, pyroptosis, ferroptosis, and entosis, as well as newly discovered cell death mechanisms such as methuosis and parthanatos may provide insights into the host-pathogen relationship and provide new therapeutic targets. Knowledge on the molecular mechanisms leading to cellular death and the outcomes of cellular death related to inflammation and immune responses, we will be better able to identify novel pathways of cellular dysfunction and achieve a deeper understanding of many pathological conditions that affect human health. In the reviewer’s opinion, the manuscript is almost ready for publication. There are only minimal typing errors.
- Line 37, “Entosis” replace with “entosis”.
- In the most section titles, the authors start the titles with the capital letter for the first word, and other words are lowercase. However, in some cases all words are wrotten in lowercase or begin with uppercase letters. This should be codified according to the principles outlined in the guide for authors.
Author Response
We thank reviewers for their valuable comments and inputs for the manuscript. We have incorporated all the changes suggested by the reviewers and revised the manuscript. Here is a point-by-point response to the reviewers’ comments and concerns.
Reviewer 1:
The manuscript is a very interesting and well-written review article presenting the current knowledge on regulated cell death responses in viral infection. Knowledge presented by the authors is necessary and useful for basic scientists as well clinicians. Molecular mechanisms underlying apoptosis inhibition or activation needs to be better understood. Research into the mechanisms, cellular, receptors, and/or microbial factors involved in the modulation of apoptosis, necroptosis, pyroptosis, ferroptosis, and entosis, as well as newly discovered cell death mechanisms such as methuosis and parthanatos may provide insights into the host-pathogen relationship and provide new therapeutic targets. Knowledge on the molecular mechanisms leading to cellular death and the outcomes of cellular death related to inflammation and immune responses, we will be better able to identify novel pathways of cellular dysfunction and achieve a deeper understanding of many pathological conditions that affect human health. In the reviewer’s opinion, the manuscript is almost ready for publication. There are only minimal typing errors.
Author Response:
We thank the reviewers for recommending our manuscript for publication. We revised the manuscript and rectified all tying errors in the revised version.
Line 37, “Entosis” replace with “entosis”.
Author Response:
We corrected Entosis” replaced with “entosis”.
In the most section titles, the authors start the titles with the capital letter for the first word, and other words are lowercase. However, in some cases all words are wrotten in lowercase or begin with uppercase letters. This should be codified according to the principles outlined in the guide for authors.
Author Response:
We thank the reviewer for pointing out our mistakes in the manuscript. We rectified those issues as per the journal guidelines.

Reviewer 2 Report
In this work, the authors review how viruses and viral infections have helped to understand the mechanisms of necrosis, pyroptosis, ferroptosis, entosis, metuosis, parthanatos. The authors should correct small writing errors in the text (uppercase, lowercase).
1- The abstract should be expressed in the past simple.
2- line 37, lowercase Entosis
3-line 39: The authors could describe in more detail the terms NETosis and ETosis
4-lines 102, 257, 322, 327, 335, 362, 368…372, 411, 649, 709 include the reference correctly.
5- It would be very interesting if the authors could make diagrams or figures that include all the genes and actors involved in each cell death described to have an overview.
6- line 573, 597. It is necessary to put the references
7- The figures should be self-explanatory and include the meaning of the abbreviations.
8- Abbreviations should be avoided in sections.
9-Explain the involvement of viruses in Entosis.
10- Authors are recommended to define each type of cell death by explaining the most critical events in a table or figure. For further details, references can be made.
Author Response
We thank reviewers for their valuable comments and inputs for the manuscript. We have incorporated all the changes suggested by the reviewers and revised the manuscript. Here is a point-by-point response to the reviewers’ comments and concerns.
Reviewer 2:
In this work, the authors review how viruses and viral infections have helped to understand the mechanisms of necrosis, pyroptosis, ferroptosis, entosis, metuosis, parthanatos. The authors should correct small writing errors in the text (uppercase, lowercase).
Author Response:
We thank the reviewer’s for pointing out the issues in the manuscript. We revised the manuscript and rectified all tying errors in the revised version.
1- The abstract should be expressed in the past simple.
Author Response:
We thank the reviewer for pointing out the issue in the abstract. We modified the abstract in the revised version of the manuscript.
2- line 37, lowercase Entosis
Author Response:
We corrected Entosis” replaced it with “entosis”.
3-line 39: The authors could describe in more detail the terms NETosis and ETosis
Author Response:
As per reviewer instruction, we described the term NETosis and Etosis.
4-lines 102, 257, 322, 327, 335, 362, 368…372, 411, 649, 709 include the reference correctly.
Author Response:
We apologise for the mistake; we corrected those references as per journal guidelines.
5- It would be very interesting if the authors could make diagrams or figures that include all the genes and actors involved in each cell death described to have an overview.
Author Response:
We thank the reviewer for the suggestion. An overview of genes involved in cell death (apoptosis, necroptosis, and pyroptosis) for DNA and RNA virus-mediated mechanisms are described in the manuscript (figure 2, 3 & 4). As per the reviewer’s instruction, we also incorporated a table (Table 1) in the revised version of the manuscript, summarizing the different types of cell death and the major regulatory mechanisms involved during viral infections.
6- line 573, 597. It is necessary to put the references
Author Response:
Yes, it’s required because the virus-induced pyroptosis is not only restricted to immune cells such as macrophages and monocytes but can also occur in epithelial cells. These references mainly address the virus-induced pyroptosis mechanisms in epithelial cells.
7- The figures should be self-explanatory and include the meaning of the abbreviations.
Author Response:
We have provided the self-explanatory meaning of the abbreviation in the figure legends in the first version of the manuscript..
8- Abbreviations should be avoided in sections.
Author Response:
We thank reviewer for pointing out the issue. We removed redundant abbreviations in the revised version of the manuscript.
9-Explain the involvement of viruses in Entosis.
Author Response:
We thank the reviewer for the suggestion; we provided information regarding the virus-mediated mechanism in Entosis.
10- Authors are recommended to define each type of cell death by explaining the most critical events in a table or figure. For further details, references can be made.
Author Response:
As per reviewer instructions, we provided a table for each type of cell death and molecular events mediated by a virus.
